# Immune Responses Induced by a Recombinant *Lactiplantibacillus plantarum* Surface-Displaying the gD Protein of Pseudorabies Virus

**DOI:** 10.3390/v16081189

**Published:** 2024-07-24

**Authors:** Assad Moon, Jingshan Huang, Xin Song, Tao Wang, Yanjin Wang, Yongfeng Li, Yuan Sun, Hongxia Wu, Huaji Qiu

**Affiliations:** State Key Laboratory for Animal Disease Control and Prevention, Harbin Veterinary Research Institute, Graduate School Chinese Academy of Agricultural Sciences, Harbin 150069, China; 2021y90100029@caas.cn (A.M.); hjs45333@163.com (J.H.); 82101231361@caas.cn (X.S.); wangtao07@caas.cn (T.W.); wangyanjin1996@126.com (Y.W.); liyongfeng@caas.cn (Y.L.); sunyuan@caas.cn (Y.S.)

**Keywords:** pseudorabies, lactic acid bacteria, probiotics, immune response, surface display

## Abstract

Pseudorabies virus (PRV) is one of the herpes viruses that can infect a wide range of animals including pigs, cattle, sheep, mice, and wild animals. PRV is a neurotropic alphaherpesvirus capable of infecting a variety of mammals. There is a rising interest in the targeted application of probiotic bacteria to prevent viral diseases, including PRV. In this study, the surface expression of enhanced green fluorescent protein (EGFP) on recombinant *Lactiplantibacillus plantarum* NC8 (rNC8) through the LP3065 LPxTG motif of *Lactobacillus plantarum* WCFS1 was generated. The surface expression was observed through confocal microscopy. Dendritic cell targeting peptides (DCpep) were also fused with LPxTG that help to bind with mouse DCs. The PRV-gD was cloned in LP3065 LPxTG, resulting in the generation of rNC8-LP3065-gD. Inactivated rNC8-LP3065-gD was administered intravenously in mice on days 1 and 7 at a dose of 200 µL (10^9^ CFU/mouse) for monitoring immunogenicity. Subsequently, a challenge dose of PRV TJ (10^4^ TCID_50_) was administered intramuscularly at 14 days post-immunization. The survival rate of the immunized mice reached 80% (4/5) with no significant signs of illness. A significant rise in anti-gD antibodies was detected in the immunized mice by ELISA. Quantitative PCR (qPCR) results showed decreased viral loading in different body tissues. Flow cytometry of lymphocytes derived from mice spleen indicated an increase in CD3^+^CD4^+^ T cells, but CD3^+^CD8^+^ T cells were not detected. Moreover, it offers a model to delineate immune correlates with rNC8-induced immunity against swine viral diseases.

## 1. Introduction

Pseudorabies virus (PRV), also referred to as Aujeszky’s disease virus or suid alphaherpesvirus 1, belongs to the *Varicellovirus* genus within the *Alphaherpesvirinae* subfamily of the *Herpesviridae*. PRV is a type of herpesvirus capable of infecting animals like pigs, cattle, sheep, mice, and other wild animals, leading to symptoms such as fever, itching (excluding pigs), and encephalomyelitis [1]. Pigs are the natural host of PRV. PRV infection in pigs can cause nervous system disorders, respiratory disease, abortions in pregnant sows, and the deaths of piglets, resulting in huge economic losses to the pig industry [2]. PRV was first reported in Chinese swine herds in 1956. Since 2012, multiple variants have caused substantial economic losses in farms. Notably, PRV can transmit from pigs to humans, causing neurological dysfunction and potentially fatal outcomes [3]. PRV is widespread worldwide, posing significant risks to both livestock and human health. Vaccination is crucial for preventing infectious diseases due to its effectiveness and cost-efficiency. However, developing safe and effective PRV vaccines is challenging due to the virus’s ability to evade immune responses and its latent, asymptomatic nature [4]. The PRV genome encodes at least 16 membrane proteins [5,6]. Up to now, various glycoproteins of PRV, such as gE, gI, gG, gB, gK, gL, gM, gC, gD, gH, and gN have been well recognized and their detailed functions and coding genes have been identified and sequenced [7]. During entry, the glycoproteins gD, gC, gB, gH, and gL are involved in virion attachment to the host cell surface resulting in the fusion of the viral envelope with the plasma membrane of host cells. gD, gL, gK, gH, and gB are essential while others are regarded as non-essential glycoproteins [8]. Envelope gD of PRV is essential for penetration but is not required for cell-to-cell spread [9]. The gB, gC, and gD can induce neutralizing antibodies, and immunization with gD can protect swine and other animals from challenges with wild-type PRV [10,11].

By modification of the vaccination route, we can have an impact on the quality of the cellular response of the host. Recently, it has been suggested that inadvertent intravenous injection of the adenovirus vector-based COVID-19 vaccine may induce platelet-adenovirus aggregates that are taken up by the spleen. This leads to a B-cell response that binds antibodies to platelets [12,13]. An intravenous (I.V.) mode of administration involves injecting the vaccine directly into the bloodstream, allowing for rapid and systemic distribution throughout the body. By bypassing the digestive system, the vaccine reaches the circulatory system quickly, triggering a robust immune response [14]. The researchers also found that there was a striking difference in the phenotypic and transcriptional quality of the CD8^+^ T-cell response in the I.V. immunization regime compared to subcutaneous vaccination [15]. Researchers have also established a vaccine against malaria administered through the I.V. route that induced superior immunogenicity and protective efficacy as compared with those of subcutaneous or intradermal administration [16].

I.V. delivery is particularly advantageous due to its precision and ability to elicit a strong immune response, leading to the production of antibodies dealing with viral infections that require a swift and comprehensive defense mechanism [17]. Many researchers have shown that I.V. administration of vaccines, such as SNP-7/8a, can alter the quality of neoantigen-specific CD8^+^ T-cell activities [18]. Additional research has shown how the dosage and route of SNP-7/8a vaccination can influence the magnitude and transcriptional quality of neo-antigen-specific T cells in the bloodstream [15].

In the past few decades, probiotics have been regarded as a safe method for managing viral diseases [19]. The antiviral activity of probiotic lactic acid bacteria (LAB) has gained widespread recognition, attributed to their ability to interact with viruses and modulate the innate immune response [20]. *L. plantarum* NC8 (NC8) has been engineered in various studies to carry viral antigens including the rabies virus, porcine epidemic diarrhea virus and Newcastle disease virus [21,22,23]. As LAB are naturally found on mucosal surfaces, particularly in the gastrointestinal tract, they are considered to be promising vehicles for the delivery of protective antigens. In particular, NC8 has attracted attention for its superior efficacy as a shuttle vector for delivery of these antigens [21].

LAB has served as a carrier for administering vaccines targeting various diseases, including ASF [24]. NC8 has significant potential for the expression of exogenous proteins. This makes it particularly promising as a vehicle for vaccines, given its ability to express a wide range of antigenic proteins [25]. NC8 exerts an immune-regulating impact by binding to the mucosal membranes of mammals’ respiratory and gastrointestinal tracts [21]. NC8 holds promise as a carrier for transporting vaccine molecules, influencing the systemic immune system. To induce effective immunity, the presence of signal peptides and anchoring proteins is crucial for securely attaching the antigens to the bacterial surface [26].

There are two primary ways to anchor the proteins to the Gram-positive bacterial cell wall: through covalent binding and non-covalent binding. The covalent attachment involves a specific motif cleaved by a sortase enzyme produced by the cell wall of the bacteria, marked by the C-terminal consensus sequence “LPxTG”, followed by hydrophobic amino acids and a short tail of positively charged amino acid residues [27]. Upon secretion, a transpeptidase known as sortase performs a cleavage between the threonine (Thr) and glycine (Gly) residues within the LPxTG motif, leading to the protein’s covalent attachment to the peptidoglycan of the cell wall through the threonine residue [27].

This study is aimed at developing a recombinant NC8 (rNC8) for surface expression of the gD antigen of PRV virus encoded by *US6* gene. To enhance the bioavailability of the targeted immunogen, LPxTG surface-anchoring protein along with dendritic cell-targeting peptide (DCpep) (AYYKTASLAPAE) were cloned into the pMG36e vector to target the dendritic cells (DCs). This study evaluated the cellular and humoral immune responses and protective efficacy of rNC8-LP3065-gD through I.V. administration.

## 2. Materials and Methods

### 2.1. Bacterial Strains, Viruses, Plasmids, and Cells

The NC8 strain and pMG36e expression plasmid were kindly provided by Prof. Guangxing Li, Northeast Agricultural University, and PRV TJ (GenBank accession number: KJ789182.1), and the corresponding PRV gene sequences were codon-optimized according to the complete genome sequence for expression in NC8 and sent to GenScript for synthesis. T cells from mice spleens were used for the evaluation of cellular immune responses. The JAWS II cells (ATCC CRL-33612) are a passaged cell line of mice DCs, cultured in MEM α (catalog no. 12571063; ThermoFisher Scientific, Beijing, China) with 25% FBS and 5 ng/mL GM-CSF at 37 °C.

### 2.2. Construction of the pMG36e-LPxTG Recombinant Plasmid

The sequences of the LPxTG motif containing proteins (LP3065, LP1447, and LP2940) were obtained from the NCBI GenBank, sourced from the complete genome of *Lactobacillus plantarum* WCFS1 (GenBank accession number: AL935263.2), and were synthesized from Rui Biotechnology. LPxTG gene fragments were inserted into the pMG36e vector between *Xba*I and *Hin*dIII. Once the confirmation of the pMG36e-LPxTG anchor was established, the vector underwent a subsequent round of digestion at the open reading frame (ORF) of the LPxTG anchor using *Xho*I and *Bam*HI restriction enzymes, and at the inserted sites of each LPxTG. Following this, the *EGFP* gene was amplified through specific primers shown in Table 1 and ligated between *Xho*I and *Bam*HI using a homologous recombination enzyme. Moreover, 6 × His-tag (5′-CACCACCACCACCACCAC-3′) and Strep-tag (5′-TGGTCCCACCCCCAGTTCGAGAAG-3′) were incorporated into the LPxTG sequence to facilitate the detection of the target protein using anti-His-tag and anti-Strep tag antibodies (Figure 1a).

The initial PCR process yielded fragments of genes: *EGFP*, and the target peptide (DCpep), along with His-tag and Strep-tag, were obtained. The LP3065-EGFP, LP2940-EGFP, and LP1447-EGFP antigens were merged with the linearized pMG36e vector, resulting in a recombinant plasmid. This plasmid was subsequently introduced into competent DH5*α* cells and incubated at 37 °C overnight until monoclonal growth occurred. The verification of the plasmids was conducted by examining the base pair length of the products obtained from colony PCR. After verification through sequencing, the plasmids were transformed into NC8-competent cells to foster the growth of clonal populations of rNC8-EGFP. The clones of rNC8-LP3065-EGFP, rNC8-2940-EGFP, rNC8-LP1447-EGFP, and rNC8-EGFP were picked and cultured in de Man–Rogosa–Sharpe (MRS) media and preserved for further experimentation.

### 2.3. Generation of the Recombinant pMG36e-NC8-PRV Plasmids

The PRV *US6* antigen encoding the gD protein was amplified through specific gene primers as shown in Table 1. The pMG36e vector was digested with *Xho*I and *Bam*HI restriction sites, and the *US6* gene fragment was ligated with digested vector using homologous recombination enzyme. The ligated plasmid was transformed into DH5*α* competent cells and cultured over LB agar and incubated at 37 °C overnight for clonal growth. The next day, clones were picked and confirmed by sequencing. The plasmid from the verified clone was extracted and electro-transformed in NC8 competent cells, cultured over MRS agar, and incubated at 37 °C clonal growth. The clones of rNC8-LP3065-gD were picked and preserved at −80 °C for further use.

### 2.4. Flow Cytometry

Flow cytometry was performed for the evaluation of the attachment of rNC8-LPxTG-EGFP strains with mice DCs. JAWS II (ATCC CRL-33612) mice DCs were cultured in cell culturing media overnight. The next day, 10^6^ cells were harvested and washed twice with phosphate-buffered saline (PBS). The recombinant strains were mixed with DCs at ratios of (1:100, 1:200 and 1:300) and incubated at room temperature for 30 min. Finally, the samples were processed using flow cytometry (ApogeeFlow System, Northwood, UK). The data were analyzed by using the Apogee histogram software v6.0.96.

### 2.5. Immunofluorescence Assay (IFA)

For IFA, rNC8-LP3065-gD strain, and rNC8-pMG36e negative control were cultured in MRS overnight and a pallet was collected by centrifugation. The pallet was washed twice with PBS and fixed using 4% paraformaldehyde for 30 min at room temperature. The pallet was washed thrice with PBS, and 1% BSA was added and incubated at room temperature for 30 min. Again, the pallet was washed, and anti-His tag monoclonal antibody (catalog no. K200060M, Beijing, China) was added and incubate at 37 °C on a rotary stirrer for 1 h. The pallet was again washed twice with PBS and secondary antibody Alexa Fluor 488 goat anti-mouse IgG (H+L) (catalog no. A-11001; ThermoFisher Scientific, Beijing, China) was added and rNC8-LP3065-gD bacteria was incubated at 37 °C for 45 min without exposure to light. After incubation, 2 µL of bacterial suspension was dropped on a glass slide, evenly applied with the side of the tip, and dried naturally and observed under a confocal microscope.

### 2.6. Western Blotting Analysis of the Recombinant Protein Expressed by rNC8-LP3065-EGFP

The recombinant strains (bearing the respective plasmids) were sub-cultured anaerobically at 37 °C overnight in an MRS liquid medium containing 10 μg/mL erythromycin. The culture that was left overnight was separated by centrifuging at 5000× *g* for 10 min at a temperature of 4 °C. The pellet was washed twice with cold PBS. A 100 μL lysozyme was added to the freshly centrifuged pellet of bacteria to break the cell wall, followed by a freeze-thaw cycle at −30 °C for 30 min. After thawing, sonication was performed to lyse the cell wall, followed by centrifugation at 10,000× *g* for 10 min. 40 μL of supernatant and 40 μL of pallet containing antigen proteins were combined with 10 μL of 5 × SDS buffer, denatured by boiling at 100 °C for 15 min, and then centrifuged at 8000× *g* for 2 min. A total of 30 μL of the protein sample was loaded onto an SDS-PAGE gel containing a 5% stacking gel and a 10% separation gel. The rNC8-LP3065-EGFP, rNC8-1447-EGFP, rNC8-LP2940-EGFP, rNC8-EGFP, and rNC8-LP3065-gD bacterial proteins were transferred to the PVDF membrane (Millipore, Darmstadt, Germany) using electro-transfer after SDS-PAGE. After the transfer was complete, the PVDF membrane was blocked using 1% skim milk solution for 3 h at room temperature. First, the PVDF membranes were incubated with the anti-EGFP and anti-His tag primary antibodies for rNC8-LP3065-EGFP, rNC8-LP1447-EGFP, rNC8-LP2940-EGFP, rNC8-EGFP, and rNC8-LP3065-gD, respectively, then incubated with IRDye 800CW goat anti-rabbit IgG Secondary Antibody for EGFP (catalog no. C40916-01, Li-Cor, Lincoln, NE, USA), and IRDye 680RD goat anti-mouse IgG secondary antibody (catalog no. C50113-06, Li-Cor, USA) for gD detection. After washing with PBST, the PVDF membranes were visualized by an Odyssey two-color infrared fluorescence imaging system (Li-Cor).

### 2.7. Immunization Protocols in Mice

Forty-four six-week-old mice, devoid of specific pathogens, were randomly distributed into four groups, with thirteen mice in groups 1, 2, and 3, and five mice in group 4 (Table 2). Subsequently, rNC8-LP3065-gD and NC8 bacterial pellets were collected from overnight cultures and underwent two washes with sterile PBS. The ultimate pellets were then re-suspended in sterile PBS, achieving an appropriate concentration for the immunization of mice. The inactivation of rNC8-LP3065-gD was carried out by putting the bacterial pallet in boiling water for 10 min. The immunization regime involved I.V. injection of 200 µL of inactivated rNC8-LP3065-gD or rNC8-pMG36e bacteria (10^9^ CFU/mice), or PBS into each mouse. The entire immunization process was carried out on day 1 as the first immunization and on day 7 as a second immunization. Two days before the challenge, five mice were euthanized, and spleens were collected for flow cytometry. On the 14th day following the booster immunization, all the mice from groups 1 to 3 were challenged with 10^4^ TCID_50_ of the PRV TJ strain via intramuscular injection to assess the protection conferred by the vaccine. After 48 h of challenge, three mice were euthanized for qPCR. The survival rates and changes in body weight among the immunized mice following the PRV challenge were monitored in other five mice and were euthanized on the 20th day after immunization. The serum samples were sequentially collected at 0, 3, 10, and 20 days post-immunization (dpi), and stored at −30 °C until use.

### 2.8. Enzyme-Linked Immunosorbent Assay (ELISA)

Specific gD-IgG antibodies from the serum of the immunized mice were detected using ELISA. The purified gD protein was coated on 96-well cell culture plates overnight at 4 °C. The next day, the plates were blocked with 0.5% skim milk for 2 h. After incubation, the plates were washed three times with 100 µL of PBST, and diluted sera (1:200) were added into each well and incubated for 1 h. The plates were washed with 100 µL of PBST thrice and Rabbit anti-Mouse IgG (H+L) (catalog no. ab6728; Abcam Shanghai, China) secondary antibodies were added, and the plates were incubated for 45 min. TMB color developer was added after washing with PBST thrice and the reaction was stopped by 2 M H_2_SO_4_. The (OD_450nm_) value of each well was detected with a microplate reader (BioTek Instruments, Inc., Winooski, VT, USA) at 450 nm.

### 2.9. Isolation of T Cells from the Spleens of the Immunized Mice

On a meticulously clean surface, the freshly extracted T cells from the spleens of the immunized mice were harvested from immunized mice groups and minced in a 35 mm sterile dish containing a 200-mesh steel mesh along with 1 mL of RPMI 1640 medium (catalog no. R8758; Sigma-Aldrich, Beijing, China). The splenic T cells were separated via a mouse spleen lymphocyte isolation kit (catalog no. CB6310; G-Clone, Beijing, China). The cell mixture underwent two washes with RPMI 1640 medium, and the supernatant was removed. An erythrocyte lysate buffer (catalog no. BL503B; Biosharp Life Sciences, Hefei, China) was added to eliminate red blood cells. Following further washing with PBS, the T cell count was determined through a cell counting machine.

### 2.10. Analysis of CD4^+^ and CD8^+^ T Cells by Flow Cytometry

Flow cytometry was utilized to assess the presence of costimulatory molecules on T cells in mice. In brief, a single-cell splenocyte suspension was gathered from the immunized mice and subsequently diluted to a concentration of 10^6^ cells/mL. A portion of this suspension was then exposed to phycoerythrin (PE)/cyanine 5 anti-mouse CD3 (catalog no. 100273), APC/cyanine 7 anti-mouse CD8a (catalog no. 100713; BioLegend), San Diego, CA, USA) and FITC rat anti-mouse CD4 (catalog no. 557307; BD-Biosciences, Beijing, China) antibodies to mark T cells (1:100 dilution), while another portion was retained without antibody treatment for negative control. Subsequently, the samples underwent a 30-min incubation at 4 °C, followed by washing twice with PBS. The cell suspension was filtered through a 5 mL Falcon 5 mL round-bottom polystyrene test tube, with a cell strainer snap cap (catalog no. 352235; Corning Science, New York, NY, USA). Finally, the samples were processed using flow cytometry (ApogeeFlow System, Northwood, UK). The data were analyzed using the Apogee histogram software v 6.0.96.

### 2.11. Measurement of PRV Genomic Copies in the Challenged Mice by Quantitative PCR (qPCR)

For analysis of viral DNA levels in challenged mice organs, tissue samples (i.e., brain, spinal cord, heart, lung, spleen, sciatic nerve, and liver) were collected and 0.1 g of each tissue sample, and 100 μL of DMEM, was added and homogenized using a TissueLyser II (Beijing, China). For quantification of viral DNA, qPCR was used to quantify PRV copies using the extracted DNA samples from mice organs as templates. Primers were designed using SnapGene (version 6.0.2, Dotmatics, Boston, MA, USA) for PRV TJ. The primers used in this study are listed in Table 1. For each sample, three replicates were performed.

### 2.12. Ethics Statements

All experimental procedures involving PRV antigen manipulation in this trial were carried out at Biosafety Level 2 (BSL2), at biocontainment facilities of the Harbin Veterinary Research Institute (HVRI) of the Chinese Academy of Agricultural Sciences (CAAS), and were approved by the Ministry of Agriculture and Rural Affairs, China. This trial was carried out in compliance with the requirements of the Animal Welfare Act and the guidelines for the care and use of experimental animals, approved by the Laboratory Animal Welfare Committee of the HVRI under approval number 230724-01-GR.

### 2.13. Statistical Analysis

The data were analyzed using two-way ANOVA in the GraphPad Prism 8.0.2. Statistical significance was determined at * *p* < 0.05, ** *p* < 0.01, and *** *p* < 0.001.

## 3. Results

### 3.1. LP3065 Surface-Anchoring Motif Expressed EGFP Significantly

Three recombinant plasmids were successfully constructed with LP3065, LP2940, and LP1447 surface-anchoring motifs expressing EGFP as shown in Figure 1a. The plasmids were transformed into NC8 expression bacteria to acquire recombinants named as rNC8-LP3065-EGFP, rNC8-LP1447-EGFP, rNC8-LP2940-EGFP, and rNC8-EGFP. The outcomes from confocal microscopy revealed the presence of green fluorescence on the bacterial cell surface, indicating the expression of EGFP in strains rNC8-LP3065-EGFP, rNC8-LP1447-EGFP, rNC8-LP2940-EGFP, and rNC8-EGFP as control. Among selected LPxTG, LP3065 and LP2940 expressed EGFP on the surface of the bacteria, while LP1447 expressed EGFP within the culture medium. The EGFP expression was more significant in LP3065 compared with LP2940. In contrast, the rNC8-EGFP strain expressed intra-cytoplasmic EGFP, and exhibited less fluorescence within the bacterial cells compared with the rNC8-LP3065-EGFP and rNC8-LP2940-EGFP groups displaying surface expression, which showed a higher intensity of green fluorescence (Figure 1b). Moreover, Western blotting results of rNC8-LP3065-EGFP, rNC8-LP1447-EGFP, rNC8-LP2940-EGFP, and rNC8-EGFP revealed rNC8-LP3065-EGFP has a significantly higher protein expression rate compared to counterparts (1.33, 1.18, 0 and 0.79 respectively) (Figure 1c).

### 3.2. rNC8-LP3065-EGFP Can Bind to the DCs

Flow cytometry was performed to evaluate the attachment of rNC8-LPxTG-EGFP with DCs. rNC8-LP3065-EGFP, rNC8-LP2940-EGFP, and rNC8-LP1447-EGFP were mixed with the DCs at ratios of 1:100, 1:200, and 1:300, and underwent flow cytometry. In flow cytometry analysis, rNC8-LP3065-EGFP exhibited notably higher fluorescence levels compared with both rNC8-LP2940 and rNC8-LP1447-EGFP. This heightened fluorescence intensity suggests a greater expression or activity of the recombinant strain with DCs. These findings indicate potential differences in gene expression and protein production among the constructs, and heightened the efficacy of rNC8-LP3065-EGFP in expressing the recombinant protein (Figure 2).

### 3.3. The Extracellular Domain of gD Was Surface-Expressed in rNC8-LP3065-gD

The recombinant plasmid pMG36e, with LP3065 anchored with gD antigen, was constructed for fusion protein expression (Figure 3a). The constructed rNC8-LP3065-gD strain was cultured and treated with primary and secondary antibodies. The IFA indicated the surface expression of the gD protein on rNC8 bacteria while no fluorescence was observed in NC8 negative control (Figure 3b).

### 3.4. Protein Expression of gD Antigen Anchored with LP3065 on the Surface of rNC8

The gD protein was expressed on the surface of rNC8 and assessed by Western blotting. The results verified the presence of the rNC8-LP3065-gD (56 kDa) protein band, which was found to be consistent with the size of rNC8-LP3065-gD antigen extract from recombinant strains by probing with anti-His tag antibody. On the other hand, no bands were observed in the extracts from the NC8 control containing pMG36e plasmid (Figure 3c).

**Figure 2 viruses-16-01189-f002:**
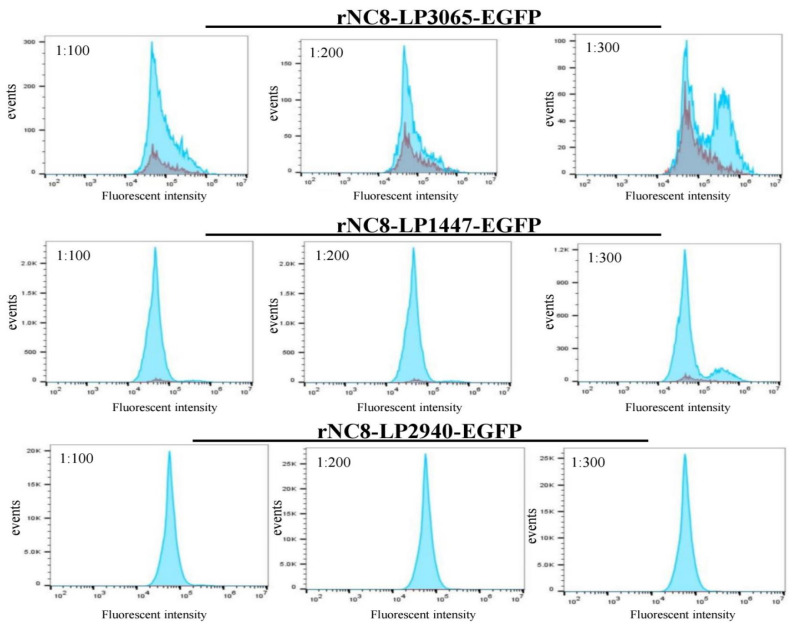
Flow cytometry of rNC8-LP3065-EGFP, rNC8-LP3065-1447-EGFP, and rNC8LP2940-EGFP binding to the DCs. The DCs were cultured overnight in a 37 °C incubator. The next day, DCs were washed twice with PBS and 10^6^ DCs were separated from the sample, then mixed with rNC8-LPxTG-EGFP at a ratio of (1:100, 1:200, and 1:300) and incubated at room temperature for 30 min. About 10^5^ cells from each sample underwent flow cytometry. The fluorescence intensity was measured that was higher in rNC8LP3065-EGFP compared with rNC8-LP1447-EGFP and rNC8-LP2940-EGFP. High fluorescence intensity showed more DCs attachment with recombinant strain.

**Figure 3 viruses-16-01189-f003:**
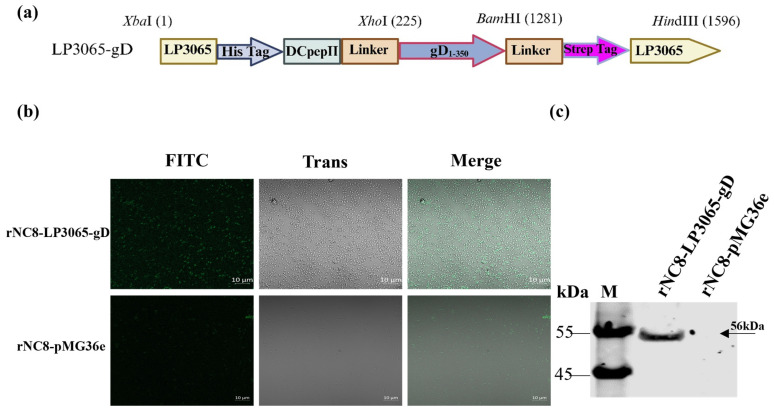
Generation and identification of rNC8-LP3065-gD. (**a**) Schematic diagram of gD protein fused with LP3065 anchoring peptide. (**b**) IFA of rNC8-LP3065-gD pallet. (**c**) Western blotting of rNC8-LP3065-gD pallet. The pallets of bacteria rNC8-LP3065 and rNC8-pMG36e were treated with RIPA lysis buffer (1:1) in volume, and then centrifuged for 5 min at 10,000× *g*, and the supernatant was used for Western blotting assay. The antibodies used for detection were anti-His Tag antibodies.

### 3.5. rNC8-LP3065-gD Induced a Rapid Production of Adaptive Immune Response in Mice Injected by I.V.

To further evaluate the role of the rNC8-LP3065-gD strain in humoral immune response, ELISA was used to evaluate the specific gD-IgG antibodies induced by the rNC8-LP3065-gD strain in mice. The mice were immunized with rNC8-LP3065-gD, rNC8-pMG36e, or PBS on days 1 and 7 at a concentration of 10^9^ CFU/mouse and the serum samples were collected on days 3 and 10 to evaluate the specific gD-IgG antibodies. The results indicate a significant rise in gD-IgG levels in group 1 compared with groups 2 and 3 (Figure 4a). The data suggest that immunization of rNC8LP3065-gD can induce humoral immune responses.

CD3^+^, CD4^+^, and CD8^+^ are the surface markers of T cells that help to differentiate between cytotoxic T cells and helper T cells. These T cells may also be effective in the promotion of NK cell activity and antigen presentation and the enhancement of macrophage lysosomal activity. Flow cytometry showed that CD3^+^CD4^+^ T cells in the spleens of mice were significantly increased in group 1 compared with groups 2 and 3 while CD3^+^CD8^+^ T cells were not detected. Depending on the degree of differentiation of the T cells, group 1 showed a higher degree of cell proliferation compared with groups 2 and 3. (Figure 4b).

### 3.6. rNC8-LP3065-gD-Immunized Mice Were Protected from PRV TJ Challenge

To evaluate the protective efficacy of the rNC8-LP3065-gD strain against PRV infection in mice post-immunization, a virus challenge experiment was conducted. On day 14, following the last vaccination, the immunized mice were intramuscularly challenged with 10^4^ TCID_50_ of the PRV TJ per mouse (Figure 5a). Four out of five immunized mice from group 1 survived after the PRV TJ challenge and showed a non-significant change in body weight compared with those in the vaccine control groups that received rNC8-pMG36e or PBS. However, each of the mice from groups 2 and 3 did not survive after challenge and died 2 days after the PRV TJ challenge (Figure 5b). The change in body weights of all the mice groups during the immunization phase remains non-significant (Figure 5c). These findings highlight the promising protective potential of I.V. immunization with the rNC8-LP3065-gD strain against PRV. To analyze the resistant effects of rNC8-LP3065-gD to the virulent strain of PRV, the PRV TJ copies in various tissues of the body were analyzed through qPCR. The tissue samples were collected from immunized groups after euthanizing the mice. The PRV TJ copies in group 1 were significantly reduced compared with groups 2 and 3. The PRV TJ copies were detected in the lungs, heart, brain, sciatic nerve, liver, spleen, and vertebral column in groups 2 and 3 while reduced copies in kidneys and legs in group 1. The data indicated that I.V. immunization with rNC8-LP3065-gD not only protects mice from a challenge but also induces a cellular and humoral immune response and reduces the viral load in body.

**Figure 4 viruses-16-01189-f004:**
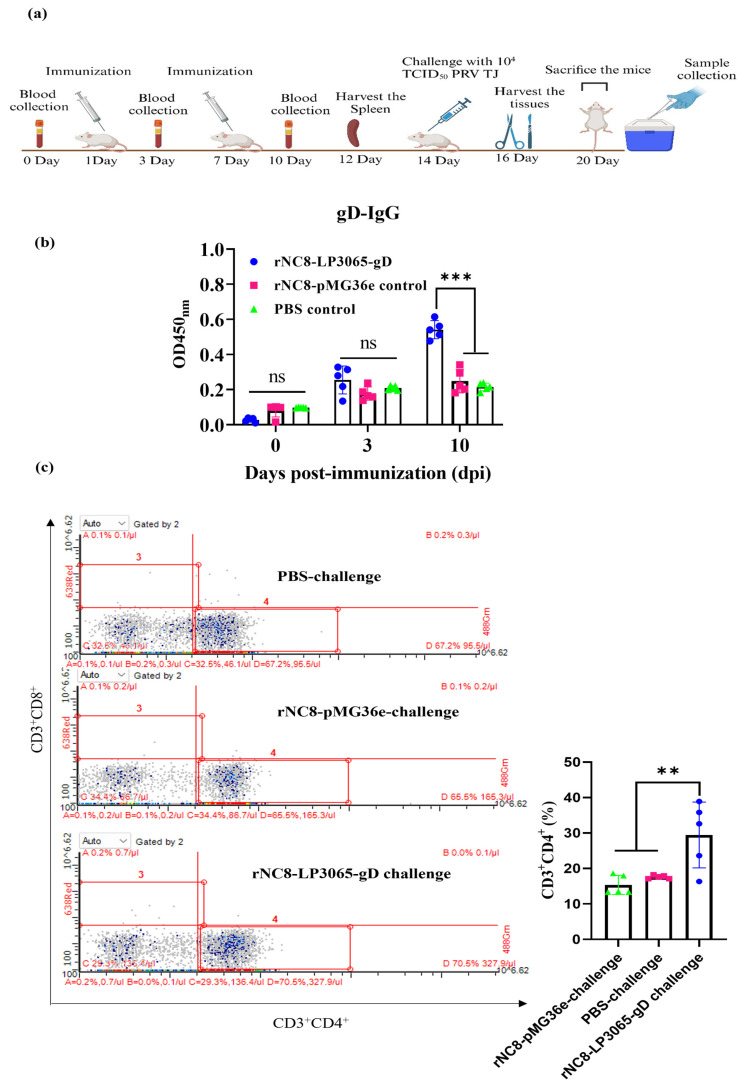
Adaptive humoral and cellular immune responses in a mouse model. The mice were immunized according to the schematic diagram. The mice were divided into four groups and immunized with 200 µL of rNC8-LP3065-gD or rNC8-pMG36e (10^9^ CFU/mouse), or PBS through the I.V. route on day 1 and 7. The specific IgG antibody production was assessed through ELISA. (**a**) Schematic diagram of immunization protocol. (**b**) IgG antibodies detection in immunized mice from groups 1 to 3. Blood samples were collected at 0, 3, and 10 dpi. Sera were separated and ELISA was performed for detection of IgG antibodies. (**c**) CD3^+^CD4^+^ T cells proliferation in mice. The spleens were harvested from the immunized groups after euthanizing the mice. Splenic T cells were collected with mouse spleen lymphocyte isolation kits and treated with anti-mouse CD3^+^, CD4^+^, and CD8^+^ antibodies at 4 °C for 30 min. The treated cells were washed twice with PBS and underwent flow cytometry. Bars represent mean ± SDs of the independent experiments; ns = not significant, ** *p* < 0.01; *** *p* < 0.001.

**Figure 5 viruses-16-01189-f005:**
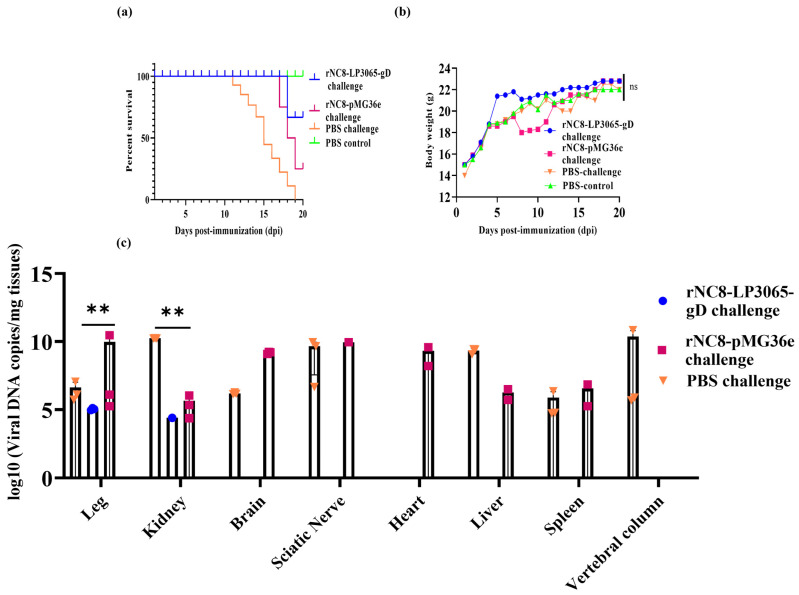
rNC8-LP3065-gD protects mice from the challenge of PRV TJ. The mice. The mice were immunized and survival rates, and the changes in body weights, were monitored during the whole regime. Viral loading was detected in the immunized mice through qPCR. A challenge dose of (10^4^ TCID_50_) was administered on day 14 and survival percentage was monitored. (**a**) Percentage survival of mice after challenge. (**b**) Body weights during immunization regime. (**c**) Viral loading in various organs after challenge with PRV TJ. The challenge dose was administered at 14 dpi. The viral DNA was extracted and the *gB* gene was quantified. Bars represent mean ± SDs of the independent experiments; ns = not significant (*p* ≥ 0.05); ** *p* < 0.01.

## 4. Discussion

The swine breeding industries have undergone rapid development in recent years. Nevertheless, infectious diseases remain the most significant challenge to the further progress of these industries. PR is an extremely contagious illness and causes significant damage to the swine production industry. PRV infections have caused serious economic losses for many countries engaged in industrialized swine production. Passive immunization through antibody mediation offers protection against invading pathogens. Hence, there is a critical need to devise innovative approaches for passive immunization to thwart PRV infections. Systemic immunity serves as the defense against infections multiplying in the body. In animals, I.V. immunization stands out for its unparalleled antigen-presetting capability, owing to its direct engagement with blood constituents and immune cells. Antigens introduced by I.V. swiftly encounter immune cells circulating in the bloodstream, triggering robust immune reactions. This route enables rapid dissemination of antigens throughout the body, eliciting robust immune responses.

I.V. administration routes necessitate less time to achieve comparable antibody levels. The mucosa harbors a rich reservoir of immune cells, enhancing antigen uptake, though the process still requires more time compared to I.V. delivery, which directly encounters the immune cells circulating in the body. Similarly, intramuscular injections require the recruitment and activation of immune cells at the injection site before mounting a systemic response. Therefore, while all routes can induce immunity, I.V. immunization stands out for its promptness in eliciting protective immune responses.

By fusing the carrier protein to an LPxTG-type cell wall anchor, proteins can be targeted to the cell wall [28]. The C-terminus consists of an LPxTG motif followed by hydrophobic amino acids and a short tail of positively charged residues. The protein anchored by LPxTG contains an N-terminal signal peptide sequence that triggers the export of the protein across the cell membrane via the secretory pathway. On release, a transpeptidase called sortase splits between the Thr and Gly in the LPxTG motif, and the protein is covalently anchored to the cell wall peptidoglycan via the threonine residue [29].

In *Lactobacillus plantarum* WCFS1, there are 26 different LPxTG motifs in the genome. In this study we used three *L. plantarum* WCFSI-originated LPxTG, i.e., LP3065, LP2940, and LP1447, with 141, 419, and 123 residues annotated as cell wall anchoring proteins. EGFP was ligated at the junctures between *Xho*I and *Bam*HI with the LPxTG motif for surface expression. Of these LPxTG-anchored proteins, only LP3065 and LP2940 expressed the EGFP protein on the cell surface, whereas LP1447 expressed the EGFP protein in the medium. This may be due to the attached signal peptide, which allows the protein to be secreted outside the medium.

Our focus primarily delved into the significance of gD antigen surface expression by probiotic bacteria in cellular and humoral immunity, such as helper T cell differentiation and IgG antibody production. Based on these required characteristics, NC8 was selected as a vaccine delivery candidate because it has been widely used in livestock as a delivery vehicle for the expression of multiple viral integral membrane proteins [23]. Probiotic-based vaccines can regulate immune responses and activate antigen-presenting cells (APCs) such as macrophages and DCs [30,31]. DCs are recognized as professional APCs. They initiate a primary immune response to pathogen challenge. In addition, NC8 is capable of high recombinant protein expression, efficient expression, and secretion of envelope proteins, and it generates potent immune protection against viruses [27].

*US6* antigen-encoding gD protein from PRV was selected for ligation with the pMG36e-LP3065 vector to anchor to the surface of NC8, because this antigen had been shown to produce antibodies and is responsible for the attachment of capsid and internalization of the virus to the cell [29]. gD protein was successfully ligated and expressed with a surface anchoring domain and proteins expressed over the rNC8 cell surface. 

Our results showed that the use of rNC8-LP3065-gD strains as antigens for vaccination in mice proved to be effective and safe. In the immunized mice, ELISA results showed that rNC8-LP3065-gD produced significant levels of IgG after a booster shot in mice from group 1 compared with groups 2 and 3 (Figure 4a). This heightened immune response suggests a potential protein to serve as an effective immunogen, warranting further investigation into its utility in vaccine development and immune modulation strategies.

On this basis, expressing rNC8-LP3065-gD enhanced protection against infection in vivo, which was confirmed by flow cytometry. CD3^+^CD4^+^ T cells orchestrate the formation and potentially maintain secondary lymphoid organs, as well as support adaptive immune responses in terms of the production of specific antibodies against invading pathogens [32]. Flow cytometry showed that the spleen of mice in group 1 showed a significant rise in CD3^+^CD4^+^ T cells, unlike in groups 2 and 3, while CD3^+^CD8^+^ T cells were not detected in any group. The expressions of CD3^+^CD4^+^ T cells in the spleen of mice in group 1 increased significantly and enhanced the immune response by producing high levels of IgG antibodies, leading to the survival of mice after a challenge with the PRV TJ. Similar findings were observed as those in previous research [33]. 

Post-immunization with the rNC8-LP3065-gD strain, all mice remained in good health and did not change in body weights and general body condition. Initially, none showed signs of disease. However, after a challenge with PRV TJ, one mouse died from group 1. These results are consistent with the previous research in which inactivated PRV (Bartha-K61) was intravenously injected with inactivated *Enterococcus faecium* [33]. We also investigated the impact of rNC8-LP3065-gD on viral loading in various organs, including the kidneys, brain, heart, lungs, spleen, spinal cord, and sciatic nerve. qPCR findings revealed a significant reduction in viral loading in group 1 mice organs compared with groups 2 and 3. Notably, rNC8-LP3065-gD effectively lowered viral presence, with the only detectable virus concentrations being observed in the legs and kidneys, albeit at low levels. This suggests a promising potential for rNC8-LP3065-gD in mitigating viral dissemination and associated pathology, particularly in critical organs.

The novelty of this system has also attracted attention. Flow cytometry, ELISA, and qPCR analysis showed that surface-expressed gD antigen on recombinant probiotic strain conferred protection against PRV in mice while being able to induce specific IgG antibodies and cellular immunity against PRV, and improved specific humoral immune responses. The focus of future studies should be to evaluate the surface-anchored recombinant rNC8 strain in pigs by ligating other viral antigens that can better induce an immune response [27].

## 5. Conclusions

We screened out the most appropriate surface anchoring protein among three selected LPxTG motif-containing proteins (LP1447. LP2940 and LP3065). The pMG36e vector was constructed with ligation of LP3065-gD antigens and transformed into NC8 resulting in rNC8-LP3065-gD. The immunization of rNC8-LP3065-gD-engineered strains protected mice from lethal challenge. The gD IgG antibodies were detected, which revealed enhanced humoral immune response, and cellular immune response was observed by increased CD3^+^CD4^+^ T cells in flow cytometry. The immunization also reduced the viral load in body tissues. Our findings imply that I.V. immunization can safely be administered to protect mice from PRV lethal challenge.

## Figures and Tables

**Figure 1 viruses-16-01189-f001:**
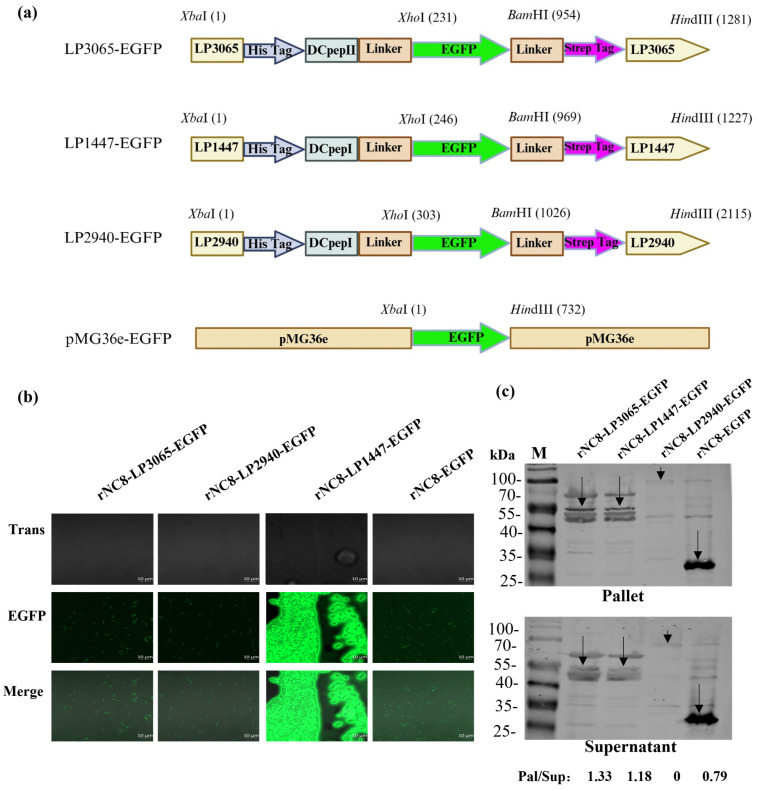
EGFP integration into pMG36e-LPxTG plasmid and surface display in rNC8. (**a**) Schematic diagram of vector construction. EGFP is ligated in between *Xho*I and *Bam*HI restriction sites. (**b**) Confocal microscopy of surface-expressed EGFP in rNC8 bacteria. For the surface expression, rNC8-LP3065-EGFP, rNC8-LP1447-EGFP, rNC8-LP2940-EGFP, and rNC8-EGFP were cultured in MRS medium with 10 µg of erythromycin overnight at 37 °C. The pallet was collected and washed twice with PBS. The pallet was resuspended in PBS, and one drop was put on the slide cover with a coverslip and observed under a confocal microscope. rNC8-LP3065-EGFP and rNC8-LP2940-EGFP expressed EGFP on the surface of NC8 bacteria, while rNC8-LP1447-EGFP secreted EGFP into the medium. The anti-EGFP antibody was used for targeted bands detection. Scale bar = 10 µm. (**c**) Western blotting analysis of rNC8-LP3065-EGFP, rNC8-LP3065-1447-EGFP, rNC8-LP2940-EGFP, and rNC8-EGFP. The arrows represent the protein bands on the PVDF membrane.

**Table 1 viruses-16-01189-t001:** Primers used in this study.

Primers	Sequence ^a^ (5′-3′)
LP1447-F	GGCGGCCGCGGCGGATCTAGAATGGCTAAATTCAGGCGTCTAGTT
LP1447-R	ACGTGCTGTAATTTGAAGCTTTCACTCCCTTCGGCGTTGCAT
LP2940-F	GGCGGCCGCGGCGGATCTAGAATGTCAAAAGCGCTTAAGATAGTGA
LP2940-R	ACGTGCTGTAATTTGAAGCTTTTAATCAGTTGTTTTATGGCGCC
LP3065-F	GGCGGCCGCGGCGGATCTAGAATGCCGAATAAATGGTGGCGATT
LP3065-R	CACGTGCTGTAATTTGAAGCTTTTACGCATTCCGTTCACCCCCAT
LPxTG-EGFP-F	GGCGGCGGAGGTTCACTCGAGATGGTGAGCAAGGGCGAGGAGC
LPxTG-EGFP-R	GGTGGGACCAGGCTGAGGATCCCTTGTACAGCTCGTCCATGCCG
pMG36e-EGFP-F	CGCCCGGGGATCGATCCTCTAGAATGGTGAGCAAGGGCGAGGAGC
pMG36e-EGFP-R	GTTTTCAGACTTTGCAAGCTTTTACTTGTACAGCTCGTCCATGCC
LP3065-gD-F	AAGGCGGCGGAGGTTCACTCGAGATGCTGCTCGCAGCGCTATT
LP3065-gD-R	GTGGGACCAGGCTGAGGATCCCGAGA GCCCGGCGCGGCGGTGGTCC
PRV-F (qPCR)	TCCTCGACGATGCAGTTGAC
PRV-R (qPCR)	ACCAACGACACCTACACCAAG

^a^ The underlined nucleotides are the positions that anneal to the vector.

**Table 2 viruses-16-01189-t002:** Immunization protocols for the mice.

Groups	Inoculum	Number of Mice	Dose per Mouse	PRV TJ Challenge
1	rNC8-LP3065-gD	13	10^9^ CFU, 200 µL	Yes
2	rNC8-pMG36e	13	10^9^ CFU, 200 µL	Yes
3	PBS	13	200 µL	Yes
4	PBS	5	200 µL	No

## Data Availability

The original contributions presented in the study are included in the article/Appendix A, further inquiries can be directed to the corresponding authors.

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
