# Peer review of "Immune Responses Induced by a Recombinant Lactiplantibacillus plantarum Surface-Displaying the gD Protein of Pseudorabies Virus"

_viruses, 2024, doi:10.3390/v16081189_

Round 1

Reviewer 1 Report

Comments and Suggestions for Authors

Major comments

This manuscript describes a bacteria-based vaccine approach against Pseudorabies virus (PRV) which can cause a serious viral disease in swine. The authors engineered a common probiotic strain Lactiplantibacillus plantarum NC8 for surface display of PRV antigen gD and a DCpep with EGFP marker and two tags(His-tag and Strep-tag). This engineered strain (rNC8-LP3065-gD) was evaluated by Microscopy, Western botting and Flow cytometry. Then, the bacteria were inactivated by boiling for 10min and then administered to mice by intravenous injection (I.V.). The virus challenge indicated that bacterial vaccine group got 85% protection (4/5). The antisera showed gD specific antibody response and CD3/CD4 positive cells were also increased. In general, it is a positive outcome from this novel vaccine approach which has provided a proof-of concept. However, directly injecting bacteria into the blood stream is not a desirable choice for swine vaccination and especially by intravenous injection which needs skilled people and will be costly.

Minor comments

1.      The statement in Introduction on page 1: “Intravenous (I.V.) vaccine delivery is a crucial and effective method in the prevention and control of viral diseases”. I do not think we have much evidence to say that from cited references 8 and 9.

2.      The LPXTG anchor-protein size and sequence from LP3605 strain should be provided and the LPXTG motif should be marked in the sequence.

3.      The DCpep sequence should be listed.

4.      In the Western blot figures (such as Fig. 1c and Fig. 3c), the antibody used (Anti-His_tag, Anti-Strep_tag or Anti-gD) should be labeled in the figures or in the figure legends.

Author Response

Responses to Reviewer #1:

Minor comments

  1. The statement in introduction on page 1: “Intravenous (I.V.) vaccine delivery is a crucial and effective method in the prevention and control of viral diseases”. I do not think we have much evidence to say that from cited references 8 and 9.

Answer: The corresponding lines have been modified. References 8 and 9 in the previous manuscript have been replaced by references 12 and 13 in the revised manuscript.

  1. Li, C.; Chen, Y.; Zhao, Y.; Lung, D.C.; Ye, Z.; Song, W.; Liu, F. F.; Cai, J. P.; Wong, W. M.; Yip, C.C. Y. Intravenous Injection of Coronavirus Disease 2019 (COVID-19) mRNA Vaccine Can Induce Acute Myopericarditis in Mouse Model. Clin. Infect. Dis. 2022, 74, 1933-1950. doi: 10.1093/cid/ciab707.
  2. Nicolai, L.; Leunig, A.; Pekayvaz, K.; Esefeld, M.; Anjum, A.; Rath, J.; Riedlinger, E.; Ehreiser, V.; Mader, M.; Eivers, L. Thrombocytopenia and Splenic Platelet-DirectedImmune Responses After IV ChAdOx1 nCov-19 Administration. Blood. 2022, 140, 478-490. doi: 10.1182/blood.2021014712.

  1. The LPXTG anchor-protein size and sequence from LP3605 strain should be provided and the LPXTG motif should be marked in the sequence.

Answer: The LP3065-gD anchor-protein size is 56 kDa mentioned in Line 349. Moreover, the LPxTG sequence is provided in the supplementary information.

  1. The DCpep sequence should be listed.

Answer: The DCpep sequence (AYYKTASLAPAE) has been updated in the manuscript in (Lines 105-106).

  1. In the Western blot figures (such as Fig. 1c and Fig. 3c), the antibody used (Anti-His tag, Anti-Strep tag or Anti-gD) should be labeled in the figures or in the figure legends.

Answer: The antibody used for detection has been updated in the original manuscript. Anti-EGFP antibody was used to detect the bands in Figure 1c in (Line 323), while anti-His Tag antibody was used for detection in Figure 3b and c in (Line 358).

Reviewer 2 Report

Comments and Suggestions for Authors

In the submitted work, authors constructed a recombinant Lactobacillus plantarum (NC8) expressing gD protein of pseudorabies virus and evaluated its immunogenicity in a mouse model. Firstly, the authors used EGFP as a reporter protein to screen LPxTG motif of Lactobacillus plantarum WCFS1. The LP3065 peptide can surface-display the EGFP protein on the cell wall of rNC8-LP3065-EGFP better than others. Secondly, the gD protein was cloned using the same vector and method as EGFP to get recombinant rNC8-LP3065-gD. Lactobacillus plantarum is a probiotics, and it is safe as a vector to express viral antigen. This is a valuable research. Finally, inactivated rNC8-LP3065-gD was administered intravenously in mice and challenged with PRV TJ. The author's innovative use of intravenous injection route immunized rNC8-LP3065-gD in mouse models. Extensive revision of the results and discussion are also needed to reach reasonable conclusions regarding the experiments performed. The language also needs to improve. These changes are summarized in the following.

Introduction

1. “US6 antigen of PRV encoding gD protein” US6 is the name of gene. The description is unclear.

2. A dendritic cell-targeting peptide (DCpep) were introduced into the pMG36e vector for the I.V. delivery system. The description is appropriate.

Materials and Methods

1. Freshly extracted T cells from mice spleens is repeated description in the 2.9

2. days post-immunization (dpi) is repeated description. 

3. The detail information of DCs should be supplied in the Material and method.

4. The primers for EGFP fused with LPxTG and pMG36e-EGFP may be different. The primers for pMG36e-EGFP are not showed in the table1.

5. The PRV antigen fragment was ligated by digesting the pMG36e-LP3065 vector with XhoI and BamHI restriction sites at 37°C with homologous recombination enzyme.” PRV antigen fragment is not a gene. It can not be ligated with a vector. Moreover, the vector is digested by homologous recombination enzyme?

6. 2.4 is not a method.

7. “Washed the pallet twice with phosphate-buffered saline (PBS) and 4% paraformaldehyde as tissue fixative was added and put at rotatory stirrer for 30 min” The description is ambiguous.

8. Move primer sequences for qPCR into Table 1

 Results

1. Figure 1 c “Wester blotting” word error.

2. The experimental goals and conclusions are not consistent.

3. Figure.3A is not described in the Result.

4. The evaluation of immune effect of commercialized PRV vaccine by intravenous injection should be supplied.

5. Figure 4B is not clear.

6. The time of Figure 5B and 5C mismatch Figure 5A.

7. Recommend moving The Figure 5A to position Figure 4A.

Discussion

The discussion section only discussed the results of this article, without comparing or discussing with other studies.

Comments on the Quality of English Language

The quality of language is fine.

Author Response

Responses to Reviewer #2:

Introduction

  1. “US6 antigen of PRV encoding gD protein” US6 is the name of gene. The description is unclear.

Answer: US6 is the gene encoding the gD protein. The statement has been updated in Line 103 in the revised manuscript.

“This study is aimed at developing a recombinant NC8 (rNC8) for surface-expression of the gD antigen of PRV virus encoded by the US6 gene”.

  1. A dendritic cell-targeting peptide (DCpep) were introduced into the pMG36e vector for the I.V. delivery system. The description is appropriate.

Answer: The statement has been updated according to the suggestion in Lines 105-106 as “To enhance the bioavailability of the targeted immunogen, LPxTG surface-anchoring protein along with dendritic cell-targeting peptide (DCpep) (AYYKTASLAPAE) were cloned into the pMG36e vector to target the dendritic cells (DCs)”.

Materials and Methods

  1. Freshly extracted T cells from mice spleens is repeated description in the 2.9.

Answer: We have deleted the repeated description in section 2.1 and modified in section 2.9 as “the freshly extracted T cells from mice spleens were harvested from immunized mice groups”.

  1. days post-immunization (dpi) is repeated description.

Answer: The repeated statement has been deleted (Lines 213-216).

  1. Detailed information of DCs should be supplied in the Materials and methods.

Answer: The mice DCs used for this study has been updated in the Materials and Methods Section 2.1 in Lines 116-118 as “The JAWS II cells (ATCC CRL-33612) are passaged cell line of mice DCs, it was cultured in MEM α (catalog no. 12571063; ThermoFisher Scientific, China) with 25% FBS and 5 ng/mL GM-CSF at 37°C”.

  1. The primers for EGFP fused with LPxTG and pMG36e-EGFP may be different. The primers for pMG36e-EGFP are not shown in the Table 1.

Answer: The primers for pMG36e-LPxTG-EGFP and pMG36e-EGFP have been updated in Table 1.

  1. “The PRV antigen fragment was ligated by digesting the pMG36e-LP3065 vector with XhoI and BamHI restriction sites at 37°C with homologous recombination enzyme.” PRV antigen fragment is not a gene. It cannot be ligated with a vector. Moreover, the vector is digested by homologous recombination enzyme?

Answer: The statement has been updated as “The pMG36e vector was digested at XhoI and BamHI restriction sites, and US6 gene fragment was ligated with digested vector using homologous recombination enzyme” in (Lines146-148).

  1. 4 is not a method.

Answer: We moved Section 2.4 into the figure legends of Figure 1b and added the flow cytometry protocols.

  1. “Washed the pallet twice with phosphate-buffered saline (PBS) and 4% paraformaldehyde as tissue fixative was added and put at rotatory stirrer for 30 min” The description is ambiguous.

Answer: The statement has been rephrased in Lines 166-167 as “The pallet was washed twice with phosphate-buffered saline (PBS) and fixed using 4% paraformaldehyde for 30 min at room temperature”.

  1. Move primer sequences for qPCR into Table 1.

Answer: The primer sequences for qPCR have been listed in Table 1.

Results

  1. Figure 1 c “Wester blotting” word error.

Answer: Corrected.

  1. The experimental goals and conclusions are not consistent.

Answer: The conclusion section has been improved as per suggestions.

  1. 3A is not described in the Result.

Answer: Figure 3a has been highlighted (Line 343).

  1. The evaluation of immune effect of commercialized PRV vaccine by intravenous injection should be supplied.

Answer: In the previous study, the same experiments of administration of inactivated PRV vaccine with a probiotic strain intravenously have been conducted. Our results are consistent with the previous findings by protecting the mice from PRV lethal challenge and increasing the survival rate up to 80%. The related study reference has been included in the Discussion section in (Lines 536-537).

  1. Figure 4B is not clear.

Answer: The figure has been updated and resized according to suggestions.

  1. The time of figure 5B and 5C mismatch Figure 5A.

Answer: Timeline for Figures 5b and c have been adjusted according to suggestions.

  1. Recommend moving Figure 5A to position Figure 4A.

Answer: The figure has been relocated to 4A.

Discussion

The discussion section only discussed the results of this article, without comparing or discussing it with other studies.

Answer: In the Discussion, the results has been modified and added in Lines 518-522, 527-531, and 536-537.

Reviewer 3 Report

Comments and Suggestions for Authors

In this research article, the authors investigate the potential of a recombinant stain to induce immunization against pseudorabies virus (PRV) in mice.

There are several drawbacks and methodological errors that should be addressed for this manuscript to be recommended for publication:

1.       The aim and significance of this study are not clearly stated in the text. Specifically, the abstract and introduction sections do not contain any information on the biological relevance of PRV, the target host, or implications in mammal health. These sections should be modified accordingly.

2.       The rationale for the selection of Lactiplantibacillus plantarum NC8 as a carrier/adjuvant is not adequately addressed in the introduction/materials and methods sections. References 18 and 19 do not contain any information on strain NC8. Additionally, to the best of my knowledge GRAS and GPS are only concerned with bacteria added in food and feed, and not for those intended for intravenous administration. The corresponding lines should be modified. The source of the strain should be included in the materials and methods section.

3.       In section 2.2 I believe the authors refer to LPxTG motif-containing proteins and not of LPxTG proteins. LPxTG is a cell wall anchor domain contained in multiple proteins, and not a protein itself.

4.       The approach followed in 2.4 does not seem appropriate to determine the localization of a protein on bacterial cells.

5.       Section 3.7: the most appropriate control would be recombinant NC8 carrying the same plasmid that rNC8-LP3065-gD had without LP3065, and not just the heat inactivated strain.

6.       Figure 2: there are significantly fewer events in the 3 graphs for rNC8-LP3065-EGFP compared to the other two recombinant strains, thus the difference described by the authors may be due to experimental error. Also, the materials and methods section does not contain information about this experimental procedure.

7.       Figure 3: I would expect no signal from the NC8-control samples.

8.       Section 3.5: as I can understand from the experimental design the authors studied specifically the presence of IgG antibodies specific to the gD protein and not total IgG counts. If so, please make the appropriate clarifications in the text.

9.       Figure 4: in the graph for CD3+CD4+ (%) cells the second column only shows results from two animals and not from the five of the group.

10.   Figure 5d: The number of animals in each column varies. Why is that?

Author Response

Responses to Reviewer #3:

  1. The aim and significance of this study are not clearly stated in the text. Specifically, the abstract and introduction sections do not contain any information on the biological relevance of PRV, the target host, or implications in mammal health. These sections should be modified accordingly.

Answer: The information regarding significance of study and biological relevance has been added in Abstract (Lines 12-14), and in the Introduction (Lines 34-45) of the revised manuscript.

  1. The rationale for the selection of Lactiplantibacillus plantarum NC8 as a carrier/adjuvant is not adequately addressed in the introduction/materials and methods sections. References 18 and 19 do not contain any information on strain NC8. Additionally, to the best of my knowledge GRAS and GPS are only concerned with bacteria added in food and feed, and not for those intended for intravenous administration. The corresponding lines should be modified. The source of the strain should be included in the materials and methods section.

Answer: The information for the use of Lactiplantibacillus plantarum NC8 as carrier has been updated in Lines 79-84 and 86-88. Moreover, the references has been updated in the revised manuscript. For IV administration, we selected this strain because this is harmless and can enhance immune system efficiently. The source of the strain has also been updated in the Materials and Methods in Lines 111-112.

  1. In section 2.2 I believe the authors refer to LPxTG motif-containing proteins and not LPxTG proteins. LPxTG is a cell wall anchor domain contained in multiple proteins, and not a protein itself.

Answer: Yes, it is LPxTG motif-containing proteins. LPxTG is the cell wall anchor motif which is the part of proteins. Line 120 has been modified.

  1. The approach followed in 2.4 does not seem appropriate to determine the localization of a protein on bacterial cells.

Answer: For the determination of protein expression on bacterial cells, we use various assays such as Western blotting, IFA and flow cytometry. One method is not enough for the determination. We just observed the EGFP fluorescence by this assay. Moreover, section 2.4 has been replaced by flow cytometry.

  1. Section 3.7: the most appropriate control would be recombinant NC8 carrying the same plasmid that rNC8-LP3065-gD had without LP3065, and not just the heat inactivated strain.

Answer: Section 3.7 does not exist in the manuscript; we have changed the negative control NC8 as rNC8-pMG36e in Figure 3b and 3c (Lines 356 and 363-364).

  1. Figure 2: there are significantly fewer events in the 3 graphs for rNC8-LP3065-EGFP compared to the other two recombinant strains, thus the difference described by the authors may be due to experimental error. Also, the materials and methods section does not contain information about this experimental procedure.

Answer: In Figure 2, the flow cytometry was conducted with similar parameters for all the samples, and the number of events depends upon the capabilities of the flow cytometry machine to analyze the sample which is automated. The difference of the fluorescence is evident between graphs. The detailed protocols for flow cytometry have been updated in the revised manuscript under Section 2.4.

  1. Figure 3: I would expect no signal from the NC8-control samples.

Answer: Figure 3 has been updated according to suggestions.

  1. Section 3.5: as I can understand from the experimental design the authors studied specifically the presence of IgG antibodies specific to the gD protein and not total IgG counts. If so, please make the appropriate clarifications in the text.

Answer: The corresponding text has been modified (Lines 221, 362, 365 and 366).

  1. Figure 4: in the graph for CD3+CD4+ (%) cells the second column only shows results from two animals and not from the five of the group.

Answer: The graph in Figure 4c has been updated.

  1. Figure 5d: The number of animals in each column varies. Why is that?

Answer: The graph in Figure 4a and 5c have been updated in the revised manuscript. We euthanized 5 animals for flow cytometry, 3 animals for qPCR and 5 animals for survivability test. Moreover, the mice experiment has been updated in the revised manuscript (Lines 210-211 and 213-216).

Figure 4a

Figure 5c
